# The Role of Monocyte Distribution Width (MDW) in the Prediction of Death in Adult Patients with Sepsis

**DOI:** 10.3390/microorganisms13020427

**Published:** 2025-02-15

**Authors:** Dimitrios Theodoridis, Angeliki Tsifi, Emmanouil Magiorkinis, Xenofon Tsamakidis, Apostolos Voulgaridis, Evgenia Moustaferi, Nikoletta Skrepetou, Sotirios Tsifis, Anastasios Ioannidis, Efstathios Chronopoulos, Stylianos Chatzipanagiotou

**Affiliations:** 1Hematology Laboratory, Konstantopoulio General Hospital, 14233 Nea Ionia, Greece; evgenia_moustaferi@yahoo.gr; 2Department of Pathophysiology, General Hospital of Athens LAIKO, 11527 Athens, Greece; atsifi@hotmail.com; 3Hematology Laboratory, Metaxa Cancer Hospital, 18537 Pireas, Greece; m.magiork@gmail.com; 4Gastroenterology Clinic, Oncology Hospital of Athens “Saint Savvas”, 11522 Athens, Greece; xenotsam@gmail.com; 5Intensive Care Unit, Konstantopoulio General Hospital, 14233 Nea Ionia, Greece; apovoulga@gmail.com; 6Hematology Department, Konstantopoulio General Hospital, 14233 Nea Ionia, Greece; nskrepetou@gmail.com; 7Department of Infectious Diseases, Fondazione IRCCS Policlinico Sa Matteo, 27100 Pavia, Italy; salnessuno@gmail.com; 8Laboratory of Basic Health Sciences, Department of Nursing, Faculty of Health Sciences, University of Peloponnese, 22100 Tripoli, Greece; tasobi@uop.gr; 9School of Medicine, National and Kapodistrian University of Athens, 11527 Athens, Greece; stathi24@yahoo.gr; 10Department of Biopathology and Clinical Microbiology, Aeginition Hospital, Medical School, National and Kapodistrian University of Athens, 11528 Athens, Greece; schatzipa@gmail.com

**Keywords:** sepsis, MDW, prognosis, death, infection

## Abstract

Sepsis is a life-threatening condition; it is a major cause of hospital mortality worldwide and it constitutes a global health problem. This research investigates the use of MDW as a predictor for septic patients. This was a double-center prospective cohort study of adult septic patients. Septic patients were identified and were categorized into two categories: those who improved and those who died. Blood was drawn from the patients three times, on the first, third, and fifth day of their admission to the hospital. MDW was evaluated as a biomarker to predict patient outcome. In addition, existing inflammatory markers were recorded in all patients. The MDW was able to predict patient’s outcome. The average MDW was found to be significantly higher in patients who died in all records. For example, an MDW value of 28.4 on the first day of admission to the hospital was shown to be the best cut-off value in determining fatal outcomes; receiver operating characteristic (ROC) analysis revealed an area under the curve value of 0.71 (95% Confidence Interval-CI: 0.57–0.84) with a sensitivity of 64.7% and a specificity of 88.2%. In conclusion, MDW, in addition to being a marker that can quickly detect sepsis more effectively than other biomarkers, which is proven by numerous studies, could also be used as an indicator to predict patient outcome. This work is an attempt in that direction.

## 1. Introduction

Sepsis, according to the Sepsis-3 conference, is a life-threatening condition characterized by the dysregulation of the host immune reaction as a response to an infection, which leads to systemic inflammation and multiple organ failure [1]. The importance of organ dysfunction has been stressed during the last decade by the creation of the Sequential Organ Failure Assessment (SOFA) score in 1994, which was employed to describe the sequence of complications of severe disease and acute patient mortality under different circumstances [2,3]. Septic shock is a serious complication of sepsis involving metabolic, cellular, and circulatory anomalies, which leads to an increased risk of mortality compared with sepsis alone [1]. It constitutes a global health problem and indicates a steady increase in incidence, with 49 million cases and 11 million sepsis-related cases.

Diagnosis and early detection of sepsis is crucial in order to improve patient survival and to reduce healthcare costs [4]. Predicting patient mortality early is also a tool that can greatly assist medical doctors in the emergency room (ER) [5]. Several biomarkers are being investigated for their reliability in predicting septic patient outcome.

Increased levels of biomarkers such as high-sensitivity C-reactive protein (hsCRP), procalcitonin (PCT), soluble urokinase-type plasminogen activator receptor (suPAR), soluble triggering receptor expressed on myeloid cells (sTREM), interleukin-6 (IL-6), lactate (LAC), and Plasminogen Activator Inhibitor-1 (PAI-1), increased levels of interleukin-8 in elderly people, and decreased levels of biomarkers such as azurocidin (AZU 1) are strongly associated with patient mortality [6,7,8,9]. 

Several other biomarkers have been explored, focusing on the parameters included in the complete blood count (CBC). The CBC is a simple exam and has several advantages, since it is a first-line test, can be easily performed, is inexpensive, quick, and available in all medical facilities. For example, an increased lower limit of red cell distribution width (RDW) is associated with sepsis mortality, and the lowest level of the platelet count in the first 3 days of admission is significantly lower in patients that die [10]. Another study showed a significant association between the basophil count and increased mortality risk [11]. Moreover, the neutrophil-to-lymphocyte ratio (NLR) and platelet-to-lymphocyte ratio (PLR) were significantly higher in septic patients who died, whichever day they were measured [12,13]. Also, studies have shown that a decreased lymphocyte count is associated with poor outcomes [14,15,16]. However, further studies on the parameters of the CBC test and sepsis mortality are essential.

Monocytes have a central role in sepsis and are a key player in the mechanisms of natural and acquired immunity. A new parameter of the CBC provided by a modern analyzer with new-generation volume–conductivity–scatter (VSC) technology is the MDW, which depicts the anisocytosis of circulating monocytes, represents the standard deviation (SD) of a set of monocyte cell volumes, and seems to be an important diagnostic and prognostic tool for the development and progression of sepsis [17]. COULTER VCS established white blood cell (WBC) leukocyte-type technology using three measurements: single cell volume, high-frequency conductivity, and laser light scattering. The combination of low-frequency current, high-frequency current, and light scattering technology provides information about each cell that can be expressed in data plots (two- and three-dimensional nephelograms, as well as surface plots).

In 2019, the Food and Drug Administration (FDA) authorized its clinical application for the detection of sepsis in adult patients at the emergency room (ER). MDW has also been tested in other clinical settings such as the intensive care unit (ICU) and the infectious disease unit, and reference intervals have been established, as well as in vitro stability tests [18,19,20,21,22,23,24,25,26,27,28]. MDW is a CBC parameter and therefore has many advantages over other biomarkers. Some of the advantages are due to the CBC test, which is widely used, low-cost, and easy to measure, does not require additional samples, has a low turn-around time, and is always available to clinicians [29]. Τhe role of MDW in prognosis and its effect on morbidity in patients has been the focus of much research, and the literature evidence strongly supports its use in clinical practice, but its role as a predictor of septic patient outcome is an area that has emerged as an interesting research question in recent years.

The main aim of our study was to investigate the role of MDW in mortality in patients with sepsis.

## 2. Materials and Methods

### 2.1. Patients and Identification of High-Risk Patients

A comparative, prospective study was carried out with 136 patients (68 patients with sepsis and 68 non-septic patients) from the Emergency Department of the General Hospital of New Ionia Konstantopouleio-Patision and Eginitio. Sepsis was defined based on the guidelines of the Third International Consensus on Sepsis and Septic Shock [1]. The Sepsis-3 definitions suggest that patients with at least two of the three clinical variables mentioned below may be prone to the poor outcome typical of sepsis: (1) low systolic blood pressure (SBP ≤ 100 mmHg), (2) high respiratory rate (≥22 breaths per min), or (3) altered mental status (Glasgow coma scale < 15). The quick SOFA (qSOFA) score includes one point for each of the above three criteria. A qSOFA score ≥ 2 with suspected infection is suggestive of sepsis or septic shock [29].

Originally, 136 patients were screened for sepsis and were divided into two groups, with 68 patients each: those with possible infection and worse prognosis and a qSOFA score ≥ 2, and those without possible infection and a qSOFA score < 2. This is how the “septic” patients came about. Patients with hematological malignancies or those undergoing recent chemotherapy or taking medications affecting monocyte population, such as injectable growth factors, were excluded from our study. Pediatric cases were also excluded, because the research hospitals do not handle pediatric cases. The patients who scored qSOFA ≥ 2 either came directly to the emergency department of the General Hospital of New Ionia Konstantopouleio-Patision or were already hospitalized in one of the two hospitals, and their clinical profile changed, resulting in them also having a qSOFA score ≥ 2. The septic patients were classified into two categories based on the Sepsis-3 classifications, “sepsis” and “septic shock”. The “sepsis” and “septic shock” patients were divided into two categories, those that improved and those that died, and we compared several biomarkers, including MDW, for these two categories.

### 2.2. Measurement of Sepsis Biomarkers

Several sepsis indicators were studied (PCT, IL-6, CRP), including the MDW of monocytes. For all patients, the following tests were performed: CBC, prothrombin time (*PT*/INR), PT-INR-activated partial thromboplastin time (*aPTT* or *APTT*), aPTT- Fibrinogen-d-dimers, serum PCT, CRP, arterial blood gas (*ABG*) and lactate (LAC), serum ferritin (FER), and serum TNF-a and *IL-6*. For the CBC and MDW calculations, blood samples were conducted in K2 EDTA vials using the Coulter DXH900 hematology analyzer (Beckman Coulter Diagnostics SA, Brea, CA, USA), and PT-INR-aPTT-FIB and d-dimers were conducted in sodium citrate vials using the BCS-XP Siemens analyzer (Siemens Healthcare Diagnostics, Hoffman Estates, IL, USA). For FER, CRP, PCT, TNF-a, and IL-6, serum was isolated from gel clot activator blood tubes; FER was measured by the chemiluminescence immunoassay using the UniCelDxI 800 Access Immunoassay System (Beckman Coulter Diagnostics SA, Brea, CA, USA), CRP by the immunoturbidimetric method using the Roche cobas c501 system (Roche diagnostics, Indianapolis, IN, USA), PCT by chemiluminescence using the Abbott Alinity C system (Abbott diagnostics, Chicago, IL, USA), and TNF-a and IL-6 by ELISA using the Brio 2 (Diachel, Attiki, Greece). LAC and ABG were measured using the ABL 800 FLEX (RADIO METER, Copenhagen, Denmark) ABG analyzer.

For each patient at sepsis, before the initiation of antimicrobial therapy, 10 mL of blood was drawn in Bactec culture vials (1 pair for each patient) and were incubated for a total of 5 days using the BD Bactec™ FX Blood system (Becton Dickinson, Franklin Lakes, NJ, USA). One blood culture set was collected from each patient, except for those for whom endocarditis was suspected, from whom three sets were collected. The biological samples were cultured and incubated in common culture media and were evaluated. Microbial isolates were identified using the Vitek 2 Compact system (Biomerieux SA, Craponne, France) and antibiograms were conducted using the MIC and the E-test method using standard criteria EUCAST.

In all patients with sepsis, the hematological markers were measured from morning samples one hour after sampling on the 1st, 3rd and 5th day in order to check their prognostic value for the patient outcome. From all patients, blood cultures were taken, as well as other biological samples such as urine, sputum, bronchoalveolar lavage, CSF, etc., in order to identify the possible source of infection before the initiation of empirical antibiotic therapy. We evaluated the clinical history of each patient, such as various comorbidities or any factor contributing to immunosuppression, co-administration of other drugs, family history, dementia, and other factors affecting the status of the patient.

### 2.3. Statistical Analysis

Quantitative variables were expressed as mean values (standard deviation) and as medians (interquartile range), while categorical variables were expressed as absolute and relative frequencies. For the comparison of proportions, chi-square tests were used. Student’s t-test was used to compare the age between the patients who died and those whose condition was improved. The Mann–Whitney test was used for the comparison of indexes between the two groups. ROC curves were used in order to estimate the predictive ability of the MMV (or MEAN-V), monocyte mean conductivity (MEAN-C), monocyte standard deviation volume (SD-V), and monocyte standard deviation conductivity (SD-C) parameters. The sensitivity and specificity were calculated for the optimal cut-offs. The area under the curve (AUC) was also calculated. All reported *p*-values are two-tailed. Statistical significance was set at *p* < 0.05, and the analyses were conducted using SPSS statistical software (version 26.0).

## 3. Results

Sixty-eight septic patients were entered in the study. The mean age was 73.4 years (SD = 16.1 years), and 54.4% were women. Half of them (*n* = 34; 50%) died. Their characteristics are presented in Table 1, for the total sample and by outcome. No significant differences were found between those who died and those whose condition was improved.

The LAC values on day 1 and 5 were significantly higher in the patients who died, as presented in Table 2. Also, the TNF-a and IL-6 values were significantly greater in the patients who died in all timepoints. An increase in TNF-a drives an excessive inflammatory response, leading to tissue damage and organ failure [30]. IL-6, a pro-inflammatory factor, increases rapidly in the serum of septic patients as it plays a crucial role in inflammatory responses, such as the activation of neutrophils or the production of immunoglobulins [31].

The patients’ mean volume (MEAN-V), mean conductivity (MEAN-C), standard deviation of volume (SD-V), and standard deviation of conductivity (SD-C) of monocytes (MONO) are presented in Table 3 by outcome. MONO SD-V is the MDW indicator, as mentioned in the introduction. It was found that the patients who died had a significantly lower MONO MEAN -C on days 3 and 5 and a significantly greater MDW on days 1, 3, and 5. Furthermore, the patients who died had a significantly greater MONO SD-C.

In the septic shock group, the patients who died had a significantly greater ΜOΝO MEAN-V (on the 5th day), significantly lower MONO MEAN-C (on the 5th day), significantly greater MDW (in all measurements), and significantly greater MONO-SdC (on the 5th day). The results are presented in Table 4.

The predictive ability of the MONO MEAN-V, MONO MEAN-C, and MONO SD-V (MDW) between those with septic events that improved and those with septic events that died was examined via ROC curves, the results of which are presented in Table 5. The MONO MEAN-C on days 3 and 5, MONO SD-V on days 1, 3, and 5, and MONO MEAN-C on day 5 had significant predictive ability. More specifically, for the MEAN-C, the optimal cut-off on the 3rd day was 123.5 with 90% sensitivity and 80.6% specificity, and on the 5th day it was 121 with 77.8% sensitivity and 72.7% specificity. For the SD-V, the optimal cut-off on day 1 was 29.3, with 64.7% sensitivity and 88.2% specificity; on day 3, it was 28.4, with 100% sensitivity and 83.9% specificity; and on day 5, it was 29.2, with 88.9% sensitivity and 90.9% specificity. For the SD-C, the optimal point on day 5 was 10.6, with 88.9% sensitivity and 90.9% specificity. CRP and PCT had no significant predictive ability, while IL-6 had significant predictive ability on days 1, 3, and 5. The predictive ability of IL-6 on day 1 and on day 3 was significantly greater than the predictive ability of the MONO SD-V on day 1 (*p* = 0.001 and *p* < 0.001, respectively) and of the MONO MEAN-C on day 3 (*p* = 0.007 and *p* = 0.007, respectively), as well as on day 5 (*p* = 0.022 and *p* = 0.039, respectively). Also, the predictive ability of IL-6 on day 1 was significantly greater than the predictive ability of the MONO MEAN-C on day 3 (*p* = 0.024).

## 4. Discussion

Our results point out that MDW, a biomarker which can be easily measured from a common CBC test, can be used for septic patients not only to detect sepsis, but also to predict the final clinical outcome. MDW and other correlated parameters such as mono mean-V and mono mean-C mono can be easily calculated from the CBC [32]. This could be of crucial importance, since the management of sepsis patients remains a major problem in clinical practice.

Studies have shown the importance of MDW as a reliable diagnostic marker for the early detection of sepsis compared to classic biomarkers such as PCT and CRP in various patient populations [17,19,20,21,23,25,33,34,35,36,37,38,39,40,41,42,43,44,45,46,47,48,49]. A published score incorporating the Modified Early Warning Score (MEWS), neutrophil-to-lymphocyte ratio (NLR), MDW, and CRP showed that a MEWS ≥ 3 with a white blood cell (WBC) count ≥ 11 × 10^9^/L, NLR ≥ 8, and MDW ≥ 20 demonstrated the highest diagnostic accuracy in all age subgroups in detecting sepsis at an early stage [50] and suggested the incorporation of MDW along with NLR and PLR to improve sepsis scores. The early detection of sepsis is crucial, since it is connected to the early initiation of broad-spectrum antibiotics, which can be lifesaving for septic patients [51]. However, there is not much research investigating whether MDW could act as a predictor of patient outcome. This is what our research tries to focus on.

In our study, patients with sepsis and a mono Mean-C value over 123.5 at day 3 and under 121 at day 5 and a Mono-SD-V above 29.3 at day 1, 28.4 at day 3, and 29.2 at day 5 could predict a worse outcome (death) in septic patients, with 90% and 80.6%, 77.8% and 72.7%, 64.7% and 88.2%, and 100% and 83.9% sensitivity and specificity, respectively, as indicated by the ROC analysis results. Other studies on septic patients point to similar results; a study by Agnello et al. [52] which evaluated various biomarkers as predictors of mortality in the ICU, such as CRP, PCT, and LAC, in comparison to MDW showed that MDW had the highest accuracy for sepsis detection (AUC 0.755, 0.708, 0.622, and 0.840 in ROC analysis, respectively). Moreover, an increase or decrease in MDW from day 1 to 4 or 5 was significantly associated with mortality or survival, respectively [53]. In a study including 252 septic patients, were able to show that an MDW value at day 3 after admission of 26.2 could distinguish between survivors and non-survivors with a sensitivity of 77.8% and a specificity of 67.6% [54]. Another study showed that an MDW value of 24.9 could predict a fatal outcome in COVID-19 patients hospitalized in the ICU (AUC 0.69, with a sensitivity and specificity of 83% and 71%, respectively) [35]. Similar results for COVID-19 patients were published with cut-off values of 20 [55], 25.4 on admission [56], and 26.4 [57] correlated. The difference between the last and first MDW is correlated with a worse prognosis in COVID-19 patients, whereas an MDW ≥ 24.685 is correlated with a poor prognosis in COVID-19 patients [58,59].

Apart from sepsis and patient outcome, MDW has been shown to be a valuable biomarker in estimating the course of other infections or inflammatory diseases. A study of 331 patients showed the value of MDW in predicting the course of cholecystitis; a value of 21.6 could predict a prolonged hospital stay [54]. Other applications of MDW could be the prognosis of COPD exacerbations [60], of Dengue infection [61], of Still’s disease [62], and of multi-inflammatory syndrome in children [63], the detection of COVID-19 infection [64], or the differential diagnosis between COVID-19 and influenza infection [65]. Τhe use of MDW as a biomarker in estimating the course of other infections or inflammatory diseases could be an area of future research.

Overall, our results point out that MDW is an independent predictor of outcome in septic patients administered in ICUs. Therefore, it is a biomarker that may be used to individualize treatment, and could even direct the physician towards a different approach, especially for those septic patients that are expected to have a worse outcome in the ICU.

Our study has some limitations. The COVID-19 era, when the data were collected, imposed several barriers. The number of patients could have been higher, but this was difficult due to the reduced number of patients during COVID-19 when the sample was selected. Due to the small size of the study population, further larger studies should be carried out to verify our results. Also, the survey does not include pediatric cases because they are not covered by the hospitals surveyed. Τhe incidence of respiratory infections could have been higher, but due to the COVID-19 era, many patients with respiratory infections were coming to the referral hospitals and not to the two survey hospitals. Future studies need to elucidate the potential application of other clinical biomarkers for prognosis in sepsis, so that in the future a combination of biomarkers can be a direct tool for doctors to safely predict patient sepsis outcome. Ιn addition, studies with a larger sample size would be useful in the future to verify the results.

Τhe request for the conducted research was made by application No.357/4-12-19 to the General Hospital of New Ionia Konstantopouleio-Patision. Τhe hospital’s management board approved the application following the positive recommendation of the hospital’s scientific committee at its second meeting on 14 April 2020 Furthermore, by the no. 252/17-3-22 decision of the 10th meeting of the management board of the Eginitio Hospital, the conduct of this research was approved.

## Figures and Tables

**Table 1 microorganisms-13-00427-t001:** Sample characteristics in total sample and by outcome.

	Total Sample (*n* = 68; 100%)	Outcome		*p*
		Improvement (*n* = 34; 50%)	Death (*n* = 34; 50%)	
	*n* (%)	*n* (%)	*n* (%)	
Gender				
Women	37 (54.4)	16 (43.2)	21 (56.8)	0.223 +
Men	31 (45.6)	18 (58.1)	13 (41.9)	
Age (years), mean (SD)	73.4 (16.1)	74.3 (16.3)	72.4 (16.1)	0.634 ++
Comorbidities				
Diabetes	14 (20.6)	7 (50)	7 (50)	>0.999 +
Hypertension	25 (36.8)	10 (40)	15 (60)	0.209 +
Heart failure	23 (33.8)	9 (39.1)	14 (60.9)	0.200 +
COPD	10 (14.7)	3 (30)	7 (70)	0.171 +
Immunosuppression	11 (16.2)	8 (72.7)	3 (27.3)	0.100 +
Other	53 (77.9)	26 (49.1)	27 (50.9)	0.770 +

+ Pearson’s chi-square test; ++ Student’s test.

**Table 2 microorganisms-13-00427-t002:** LAC, CRP, PCT, Fibrogen, FER, TNFa, and IL-6 values by outcome. Values of *p* < 0.05 are marked in bold.

	Outcome						*p*
	Improvement (*n* = 34; 50%)			Death (*n* = 34; 50%)			
	*n*	Mean (SD)	Median (IQR)	*n*	Mean (SD)	Median (IQR)	
LAC (mmol/L)							
Day 1	34	4.15 (1.87)	4.1 (2.9–5)	34	6.38 (3.99)	6.6 (3.1–8.3)	**0.045**
Day 3	34	2.92 (1.2)	2.7 (2.2–3.4)	15	5.14 (4.88)	4.2 (1.8–4.9)	0.259
Day 5	34	1.79 (0.88)	1.8 (1.2–2.2)	15	5.03 (3.57)	5.3 (0.8–8.6)	**0.007**
CRP (mg/L)							
Day 1	34	201.44 (105.62)	169 (106–306)	34	202.91 (138.03)	170 (95–251)	0.650
Day 3	34	157.03 (79.16)	139 (90–201)	17	196.59 (124)	168 (102–280)	0.509
Day 5	34	120.12 (75.02)	90 (66–176)	16	196.06 (141.64)	199 (67–324)	0.131
PCT (ng/L)							
Day 1	34	26.69 (31.47)	11.41 (1.12–46)	34	17.69 (26.17)	2.39 (0.95–24)	0.289
Day 3	34	13.78 (19.22)	7.39 (0.89–17.8)	17	6.58 (8.97)	3.44 (0.89–8.45)	0.230
Day 5	34	9.05 (21.61)	3.79 (0.8–6.7)	14	5.38 (9.35)	0.96 (0.32–5.24)	0.318
Fibrogen (mg/dL)							
Day 1	34	402.18 (116.88)	383.5 (330–482)	34	403.35 (172.08)	364 (304–542)	0.704
Day 3	34	478.91 (88.9)	489 (412–564)	17	421.59 (115.62)	420 (351–462)	0.066
Day 5	34	443.29 (95.5)	432 (378–490)	16	446.94 (145.32)	460 (297–610)	0.868
FER (ng/mL)							
Day 1	34	399.18 (289.38)	336 (172–551)	34	455.09 (302.28)	392 (277–500)	0.317
Day 3	34	427.21 (280.17)	394 (208–490)	16	412.5 (312.43)	380.5 (176–507.5)	0.519
Day 5	34	415.91 (284.7)	339.5 (248–468)	15	362 (206.7)	333 (201–452)	0.543
TNFa (pg/mL)							
Day 1	34	77.26 (13.9)	76.15 (65.1–84.6)	34	132.45 (15.62)	135.5 (129–144)	**<0.001**
Day 3	34	16.11 (2.41)	16.15 (14.2–16.9)	12	139.75 (6.34)	142 (135.5–143)	**<0.001**
Day 5	34	4.53 (0.97)	4.4 (3.7–5.4)	9	144.56 (3.94)	145 (140–147)	**<0.001**
IL-6 (pg/mL)							
Day 1	34	22.94 (22.98)	4.15 (3.9–49.8)	34	104.12 (20.88)	108.4 (103.2–115.2)	**<0.001**
Day 3	34	18.24 (18.68)	3 (3–40.2)	12	111.8 (5.07)	113.6 (108.4–114.4)	**<0.001**
Day 5	34	71.4 (22.98)	68.55 (53.9–90)	10	109.31 (20.25)	116 (112–117.6)	**<0.001**

**Table 3 microorganisms-13-00427-t003:** MONO MEAN-V, MONO MEAN-C, MDW, and MONO Sd-C values by outcome. Values of *p* < 0.05 are marked in bold.

	Outcome						*p*
	Improvement (*n* = 34; 50%)			Death (*n* = 34; 50%)			
	*n*	Mean (SD)	Median (IQR)	*n*	Mean (SD)	Median (IQR)	
MONO MEAN-V							
Day 1	34	188.76 (12.52)	189.5 (183–195)	34	186.53 (12.08)	184 (178–195)	0.329
Day 3	31	186.32 (10.71)	187 (179–192)	10	179.4 (9.31)	179.5 (169–182)	0.091
Day 5	33	179.88 (10.2)	181 (175–184)	9	185.33 (9.73)	188 (182–194)	0.068
MONO MEAN-C							
Day 1	34	120.85 (5.26)	121.5 (117–124)	34	119.03 (9.03)	120 (118–124)	0.453
Day 3	31	120.9 (4.81)	121 (119–123)	10	124.3 (1.49)	124 (124–126)	**0.004**
Day 5	33	122.55 (3.81)	123 (120–125)	9	103.33 (22.32)	120 (83–120)	**0.025**
MONO SD-V (MDW)							
Day 1	34	26.19 (3.42)	26.01 (25.09–28.74)	34	29.75 (5.38)	30.32 (24.75–32.36)	**0.004**
Day 3	31	25.36 (3.33)	24.96 (22.6–27.47)	10	31.79 (3.49)	31.14 (29.13–33.97)	**<0.001**
Day 5	33	24.14 (3.62)	24.09 (21.21–27.01)	9	34.29 (5.4)	31.64 (30.3–38.26)	**<0.001**
MONO SD-C							
Day 1	34	12.92 (10.85)	8.52 (4.83–15.97)	34	15.71 (13.19)	8.43 (5.46–20.33)	0.297
Day 3	31	20.8 (31.83)	6.85 (5.32–20.96)	10	11.05 (2.15)	10.52 (9.49–13.76)	0.524
Day 5	33	7.46 (7.14)	5.27 (4.81–5.62)	9	30.35 (20.81)	16.33 (13.48–51.13)	**<0.001**

**Table 4 microorganisms-13-00427-t004:** MONO MEAN-V, MONO MEAN-C, MDW, and MONO Sd-C values by outcome. Values of *p* < 0.05 are marked in bold.

Septic Shock Group	Outcome				*p* Mann–Whitney Test
	Improvement		Death		
	Mean (SD)	Median (IQR)	Mean (SD)	Median (IQR)	
MOΝO MEAN-V					
1st day	187.7 (13.9)	184 (180–190)	186.6 (12.5)	184 (178–196,5)	0.867
3rd day	185.6 (9.6)	186.5 (179–195)	179.4 (9.3)	179.5 (169–182)	0.149
5th day	176.1 (9.1)	179 (172–182)	185.3 (9.7)	188 (182–194)	**0.041**
MONO MEAN-C					
1st day	122.8 (4.9)	124 (120–125)	119 (9.3)	120 (118–124)	0.137
3rd day	120.9 (6.4)	121 (120–125)	124.3 (1.5)	124 (124–126)	0.108
5th day	122.5 (4.7)	124 (120–125)	103.3 (22.3)	120 (83–120)	**0.034**
MDW					
1st day	25.6 (4.2)	25.8 (23.1–27.3)	30.2 (5.3)	30.9 (27–32,6)	**0.008**
3rd day	25 (3.4)	23.9 (21.5–27.6)	31.8 (3.5)	31.1 (29.1–34)	**<0.001**
5th day	25.5 (4.9)	27 (20.5–28.1)	34.3 (5.4)	31.6 (30.3–38.3)	**0.005**
MONO-SdC					
1st day	12.6 (9)	9.4 (5.1–16)	16.2 (13.4)	11.1 (5.4–22.5)	0.519
3rd day	15.7 (12.3)	11.6 (6–19.4)	11.1 (2.2)	10.5 (9.5–13.8)	0.815
5th day	8.5 (8.4)	5.1 (4.6–6.2)	30.4 (20.8)	16.3 (13.5–51.1)	**0.002**

**Table 5 microorganisms-13-00427-t005:** ROC analysis results.

		AUC (95% CI) +	*p*	Optimal Cut-Off	Sensitivity (%)	Specificity (%)
MONO MEAN-C	Day 3	0.81 (0.67–0.95)	0.004	>123.5	90.0	80.6
	Day 5	0.74 (0.53–0.96)	0.026	<121	77.8	72.7
MONO SD-V	Day 1	0.71 (0.57–0.84)	0.004	>29.3	64.7	88.2
	Day 3	0.92 (0.83–1.00)	<0.001	>28.4	100.0	83.9
	Day 5	0.94 (0.86–1.00)	<0.001	>29.2	88.9	90.9
MONO SD-C	Day 5	0.92 (0.82–1.00)	<0.001	>10.6	88.9	90.9
CRP (mg/L)	Day 1	0.47 (0.33–0.61)	0.650	-	-	-
	Day 3	0.56 (0.36–0.75)	0.510	-	-	-
	Day 5	0.63 (0.43–0.84)	0.132	-	-	-
PCT (ng/L)	Day 1	0.43 (0.29–0.57)	0.289	-	-	-
	Day 3	0.40 (0.24–0.55)	0.231	-	-	-
	Day 5	0.41 (0.22–0.60)	0.318	-	-	-
IL-6 (pg/mL)	Day 1	0.97 (0.91–1.00)	<0.001	>66.4	97.1	100.0
	Day 3	0.99 (0.97–1.00)	<0.001	>72	100.0	100.0
	Day 5	0.89 (0.74–1.00)	<0.001	>107.2	90.0	94.1

+ Area Under the Curve (95% CI).

## Data Availability

The original contributions presented in the study are included in the article; further inquiries can be directed to the corresponding author.

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
