# Peer review of "The Role of Monocyte Distribution Width (MDW) in the Prediction of Death in Adult Patients with Sepsis"

_microorganisms, 2025, doi:10.3390/microorganisms13020427_

Round 1
Reviewer 1 Report
Comments and Suggestions for Authors
After reviewing the submitted article, I identified some points that could be improved or deserve attention.
Page 1, Title:
The title is clear, but could be more specific by mentioning "MDW in adult patients with sepsis." I suggest making it clear that this is a prospective study.
-Add details about the type of study, such as "prospective cohort study."
Page 2, Abstract, Lines 10-15:
The accuracy of MDW as a predictor is presented without sufficient details about limitations or comparison with other markers.
-Include briefly mention limitations and comparative results from other relevant studies.
Page 3, Introduction, Lines 10-20:
Although the basic concepts of sepsis are covered, more up-to-date references on biomarkers other than MDW are lacking.
-Update the references to include studies after 2020, especially on advances in sepsis markers.
Page 5, Methods, Lines 15-30:
The exclusion criterion for pediatric patients is not fully justified.
-Explain why pediatric data were not included or indicate the limitations of this exclusion.
Page 6, Results, Tables 1 and 2:
Tables present mean values and standard deviations without discussing clinical significance.
-Add clinical interpretation of values, especially for variables such as IL-6 and TNF-a.
Page 8, Discussion, Lines 20-30:
The discussion section mentions limitations but could expand on how these influence the findings and future implications.
-Insert a paragraph detailing how limitations may impact the results and suggesting directions for future research.
Page 9, References:
Some references seem outdated or generic.
-Check the relevance of citations, replacing them with more recent studies if possible.
Author Response
- Page 1, Title:
- The title is clear, but could be more specific by mentioning "MDW in adult patients with sepsis." I suggest making it clear that this is a prospective study.
- Add details about the type of study, such as "prospective cohort study."
Response to comments:
We didn’t change the title, because our article focuses on MDW as a biomarker for predicting the outcome of the septic patient and does not deal with MDW as a biomarker of sepsis in general. Ιn the summary in the “methods” section we mention that it is a double-center prospective cohort study.
- Page 2, Abstract, Lines 10-15:
- The accuracy of MDW as a predictor is presented without sufficient details about limitations or comparison with other markers.
- Include briefly mention limitations and comparative results from other relevant studies.
Response to comments:
The predictive ability of MDW between septic events that improved and septic events that died was examined via ROC curves, the results of which are presented in table 8. Limitations of our study and thus of the predictive ability of the MDW are mentioned in the “discussion” section. Table 2 presents other biomarkers in terms of their predictive ability, except for MDW. Limitations of our study MDW are mentioned in the “discussion” section. From our research we have not found any similar work investigating the ability of MDW to predict patient’s outcome and therefore we consider our work innovative.
- Page 3, Introduction, Lines 10-20:
- Although the basic concepts of sepsis are covered, more up-to-date references on biomarkers other than MDW are lacking.
- Update the references to include studies after 2020, especially on advances in sepsis markers.
Response to comments:
In “introduction” section we refer to biomarkers used as predictive tools for the effect of sepsis. We added [62] reference.
- Page 5, Methods, Lines 15-30:
- The exclusion criterion for pediatric patients is not fully justified.
- Explain why pediatric data were not included or indicate the limitations of this exclusion.
Response to comments:
We added in “Patients and identification of high-risk patients“ section “because the research hospitals do not handle pediatric cases”. Τhis parameter does not pose any limitation to our research, as it focuses on biomarkers of sepsis prognosis exclusively in adult patients.
- Page 6, Results, Tables 1 and 2:
- Tables present mean values and standard deviations without discussing clinical significance.
- Add clinical interpretation of values, especially for variables such as IL-6 and TNF-a.
Response to comments:
Below table 1 we referred to clinical significance of IL-6 and TNF-a, adding 63 and 64 references.
- Page 8, Discussion, Lines 20-30:
- The discussion section mentions limitations but could expand on how these influence the findings and future implications.
- Insert a paragraph detailing how limitations may impact the results and suggesting directions for future research.
Response to comments:
We acknowledge in this section that the small sample size could lead to erroneous results and thus further studies with larger populations are needed to confirm our findings. Also, we include a paragraph explaining the limitations. However, we cannot know the exact implications of these limitations on our research.
- Page 9, References:
- Some references seem outdated or generic.
- Check the relevance of citations, replacing them with more recent studies if possible.
Response to comments:
We took very seriously the literature references and extracted from them data that are able to assist in highlighting the results of our research.
Reviewer 2 Report
Comments and Suggestions for Authors
The manuscript provides valuable insights into MDW as a biomarker for predicting mortality in sepsis.
My criticisms are as follows:
-
Sample Size:
- The sample size of 68 septic patients is relatively small, limiting the generalizability of the findings. You should in some way define this as preliminary and interesting reuslt thaht shloudl be better addressed in future studies with larger cohorts .
-
Exclusion Criteria:
- The exclusion of patients with hematological malignancies or recent chemotherapy could introduce bias, as these populations represent a significant subset of septic patients. Please explain
-
Comparison to Other Biomarkers:
- While MDW is evaluated, the comparison to established biomarkers such as CRP, procalcitonin, and IL-6 could be expanded. The manuscript mentions their levels but does not delve deeply into their relative prognostic value compared to MDW.
-
Outcome Definitions:
- "Improvement" and "Death" are used as outcomes, but further stratification (e.g., discharge vs. prolonged ICU stay) might provide additional insights.
-
Limitations Section:
- While the authors acknowledge the study's limitations, the impact of the COVID-19 era and its potential influence on patient recruitment, sepsis profiles, and outcomes should be discussed more comprehensively.
-
Mechanistic Insights:
- While MDW is identified as a prognostic marker, the manuscript does not explore potential mechanisms linking MDW to sepsis severity and outcomes.
-
Author Response
The manuscript provides valuable insights into MDW as a biomarker for predicting mortality in sepsis.
My criticisms are as follows:
- Sample Size:
- The sample size of 68 septic patients is relatively small, limiting the generalizability of the findings. You should in some way define this as preliminary and interesting result that shloud be better addressed in future studies with larger cohorts .
Response to comments:
We acknowledge in “Discussion” section that the small sample size could lead to erroneous results and thus further studies with larger populations are needed to confirm our findings.
- Exclusion Criteria:
- The exclusion of patients with hematological malignancies or recent chemotherapy could introduce bias, as these populations represent a significant subset of septic patients. Please explain
Response to comments:
The exclusion of patients with hematological malignancies or recent chemotherapy was made because it could affect the number of monocytes. This could affect the results of our study. Ιn the future we could focus on these categories of patients in a more direct way.
- Comparison to Other Biomarkers:
- While MDW is evaluated, the comparison to established biomarkers such as CRP, procalcitonin, and IL-6 could be expanded. The manuscript mentions their levels but does not delve deeply into their relative prognostic value compared to MDW.
Response to comments:
We added in “Results” results of predictive ability of other biomarkers.
- Outcome Definitions:
- "Improvement" and "Death" are used as outcomes, but further stratification (e.g., discharge vs. prolonged ICU stay) might provide additional insights.
Response to comments:
Our study did not focus on what you mentioned, which will be the subject of future research.
- Limitations Section:
- While the authors acknowledge the study's limitations, the impact of the COVID-19 era and its potential influence on patient recruitment, sepsis profiles, and outcomes should be discussed more comprehensively.
Response to comments:
Our study did not focus on what you mentioned, but we acknowledge in “Discussion” section that the incidence of respiratory infections could have been higher, but due to the covid era many patients with respiratory infections were coming to the referral hospitals and not to the two survey hospitals.
- Mechanistic Insights:
- While MDW is identified as a prognostic marker, the manuscript does not explore potential mechanisms linking MDW to sepsis severity and outcomes.
Response to comments:
MDW depicts the anisocytosis of circulating monocytes, represents the standard deviation (SD) of a set of monocyte cell volume, which increases rapidly during the dynamic phenomenon of sepsis.
Reviewer 3 Report
Comments and Suggestions for Authors
This manuscript investigates the role of Monocyte Distribution Width (MDW) as a prognostic marker for sepsis-related mortality. The study evaluates MDW values alongside other biomarkers in a double-center prospective cohort of adult septic patients, highlighting its potential as an accessible and cost-effective tool for predicting outcomes in septic patients. The study's innovative focus on MDW as a prognostic biomarker is novel and clinically relevant, promoting its practical integration into routine workflows. Methodologically, the study is robust, employing appropriate statistical tools such as ROC analysis to validate its findings. Additionally, the inclusion of multiple biomarkers provides a comprehensive comparison, strengthening the case for MDW's utility in sepsis research. Overall this manuscript represents a strong clinical study with robust results, while some minor concerns need to be taken care of.
The manuscript contains minor language errors, such as "deathin" in the abstract and inconsistent spacing, requiring careful editing for grammatical precision and clarity. Some sentences, particularly in the results and discussion sections, could benefit from rephrasing for conciseness and improved flow. These need to be improved for the benefits of the readers.
While the sample size is justified, it could limit generalizability, and future research with a larger, multi-center cohort is recommended. More details about the non-septic control group and their clinical characteristics would provide better context for comparisons.
Although the limitations are acknowledged, the manuscript could elaborate on the potential variability of MDW across different patient populations and healthcare settings. The discussion, while referencing other studies, could delve deeper into how MDW compares to or complements existing sepsis biomarkers.
The manuscript would benefit from a careful discussion of why an independent validation cohort is less necessary in this case than in other biomarker research projects. Addressing these minor weaknesses will enhance the clarity, impact, and relevance of the manuscript.
Author Response
- This manuscript investigates the role of Monocyte Distribution Width (MDW) as a prognostic marker for sepsis-related mortality. The study evaluates MDW values alongside other biomarkers in a double-center prospective cohort of adult septic patients, highlighting its potential as an accessible and cost-effective tool for predicting outcomes in septic patients. The study's innovative focus on MDW as a prognostic biomarker is novel and clinically relevant, promoting its practical integration into routine workflows. Methodologically, the study is robust, employing appropriate statistical tools such as ROC analysis to validate its findings. Additionally, the inclusion of multiple biomarkers provides a comprehensive comparison, strengthening the case for MDW's utility in sepsis research. Overall this manuscript represents a strong clinical study with robust results, while some minor concerns need to be taken care of.
Response to comments:
Thank yοu for the review.
- The manuscript contains minor language errors, such as "deathin" in the abstract and inconsistent spacing, requiring careful editing for grammatical precision and clarity. Some sentences, particularly in the results and discussion sections, could benefit from rephrasing for conciseness and improved flow. These need to be improved for the benefits of the readers.
Response to comments:
We carefully re-read the whole text and made some changes where necessary to improve the expression.
- While the sample size is justified, it could limit generalizability, and future research with a larger, multi-center cohort is recommended. More details about the non-septic control group and their clinical characteristics would provide better context for comparisons.
Response to comments:
We are scheduling a future research with a larger, multi-center cohort, that will be easier to carry out without the constraints of the covid-era.
- Although the limitations are acknowledged, the manuscript could elaborate on the potential variability of MDW across different patient populations and healthcare settings. The discussion, while referencing other studies, could delve deeper into how MDW compares to or complements existing sepsis biomarkers.
Response to comments:
Since MDW as a biomarker for predicting patient’s outcome is a fairly new research topic, there are not many similar studies dealing with it. Therefore, the combined use of MDW as a prognostic tool in combination with other biomarkers needs further research. Ηowever, in the “introduction” section we have added some general advantages of MDW in comparison with other biomarkers. We took information from reference 65, which we added.
- The manuscript would benefit from a careful discussion of why an independent validation cohort is less necessary in this case than in other biomarker research projects. Addressing these minor weaknesses will enhance the clarity, impact, and relevance of the manuscript.
Response to comments:
We will take your comment into account in a future research.
Round 2
Reviewer 1 Report
Comments and Suggestions for Authors
NO COMMENTS